# GRAPH RESIDUAL FLOW FOR MOLECULAR GRAPH GENERATION

## ABSTRACT

Statistical generative models for molecular graphs attract attention from many researchers from the fields of bio- and chemo-informatics. Among these models, invertible flow-based approaches are not fully explored yet. In this paper, we propose a powerful invertible flow for molecular graphs, called graph residual flow (GRF). The GRF is based on residual flows, which are known for more flexible and complex non-linear mappings than traditional coupling flows. We theoretically derive non-trivial conditions such that GRF is invertible, and present a way of keeping the entire flows invertible throughout the training and sampling. Experimental results show that a generative model based on the proposed GRF achieve comparable generation performance, with much smaller number of trainable parameters compared to the existing flow-based model.

## 1 INTRODUCTION

We propose a deep generative model for molecular graphs based on invertible functions. We especially focus on introducing an invertible function that is tuned for the use in graph structured data, which allows for flexible mappings with less number of parameters than previous invertible models for graphs.

Molecular graph generation is one of the hot trends in the graph analysis with a potential for important applications such as *in silico* new material discovery and drug candidate screening. Previous generative models for molecules deal with string representations called SMILES (e.g. Kusner et al. (2017); Gómez-Bombarelli et al. (2018)), which does not consider graph topology. Recent models such as (Jin et al., 2018; You et al., 2018; De Cao & Kipf, 2018; Madhawa et al., 2019) are able to directly handle graphs. Several researchers are investigating this topic using sophisticated statistical models such as variational autoencoders (VAEs) (Kingma & Welling, 2014), adversarial loss-based models such as generative adversarial networks (GANs) (Goodfellow et al., 2014; Radford et al., 2015), and invertible flows (Kobyzev et al., 2019) and have achieved desirable performances.

The decoders of these graph generation models generate a discrete graph-structured data from a (typically continuous) representation of a data sample, which is modeled by aforementioned statistical models. In general, it is difficult to design a decoder that balances the efficacy of the graph generation and the simplicity of the implementation and training. For example, MolGAN (De Cao & Kipf, 2018) has a relatively simple decoder but suffers from generating numerous duplicated graph samples. The state-of-the-art VAE-based models such as (Jin et al., 2018; Liu et al., 2018) have good generation performance but their decoding scheme is highly complicated and requires careful training. On the contrary, invertible flow-based statistical models (Dinh et al., 2015; Kobyzev et al., 2019) do not require training for their decoders because the decoders are simply the inverse mapping of the encoders and are known for good generation performances in image generation (Dinh et al., 2017; Kingma & Dhariwal, 2018). Liu et al. (2019) proposes an invertible-flow based graph generation model. However, their generative model is not invertible because its decoder for graph structure is not built upon invertible flows. The GraphNVP by Madhawa et al. (2019) is the seminal fully invertible-flow approach for graph generation, which successfully combines the invertible maps with the generic graph convolutional networks (GCNs, e.g Kipf & Welling (2017); Schlichtkrull et al. (2017)).

However, the coupling flow (Kobyzev et al., 2019) used in the GraphNVP has a serious drawback when applied to *sparse graphs* such as molecular graphs we are interested in. The coupling flow

requires a disjoint partitioning of the latent representation of the data (graph) in each layer. We need to design this partitioning carefully so that all the attributes of a latent representation are well *mixed* through stacks of mapping layers. However, molecular graphs are highly sparse in general: degree of each node atom is at most four (valency), and only few kind of atoms comprise the majority of the molecules (less diversity). Madhawa et al. (2019) argued that only a specific form of partitioning can lead to a desirable performance owing to sparsity: for each mapping layer, the representation of only one node is subject to update and all the other nodes are kept intact. In other words, a graph with 100 nodes requires at least 100 layers. But with the 100 layers, only one affine mapping is executed for each attribute of the latent representation. Therefore, the complexity of the mappings of GraphNVP is extremely low in contrast to the number of layer stacks. We assume that this is why the generation performance of GraphNVP is less impressive than other state-of-the-art models (Jin et al., 2018; Liu et al., 2018) in the paper.

In this paper we propose a new graph flow, called *graph residual flow (GRF)*: a novel combination of a generic GCN and recently proposed residual flows (Behrmann et al., 2019; Song et al., 2019; Chen et al., 2019). The GRF does not require partitioning of a latent vector and can *update all the node attributes in each layer*. Thus, a 100 layer-stacked flow model can apply the (non-linear) mappings 100 times for each attribute of the latent vector of the 100-node graph. We derive a theoretical guarantee of the invertibility of the GRF and introduce constraints on the GRF parameters, based on rigorous mathematical calculations. Through experiments with most popular graph generation datasets, we observe that a generative model based on the proposed GRF can achieve a generation performance comparable to the GraphNVP Madhawa et al. (2019), but with much fewer trainable parameters.

To summarize, our contributions in this paper are as follows:

- propose the graph residual flow (GRF): a novel residual flow model for graph generation that is compatible with a generic GCNs.

- prove conditions such that the GRFs are invertible and present how to keep the entire network invertible throughout the training and sampling.

- demonstrate the efficacy of the GRF-based models in generating molecular graphs; in other words, show that a generative model based on the GRF has much fewer trainable parameters compared to the GraphNVP, while still maintaining a comparable generation performance.

## 2 BACKGROUND

### 2.1 GRAPHNVP

We first describe the GraphNVP (Madhawa et al., 2019), the first fully invertible model for chemical graph generation, as a baseline. We simultaneously introduce the necessary notations for graph generative models.

We use the notation $G = (A, X)$ to represent a graph $G$ comprising an adjacency tensor $A$ and a feature matrix $X$. Let $N$ be the number of nodes in the graph, $M$ be the number of the types of nodes, and $R$ be the number of the types of edges. Then, $A \in \{0, 1\}^{N \times N \times R}$ and $X \in \{0, 1\}^{N \times M}$. In the case of molecular graphs, $G = (A, X)$ represents a molecule with $R$ types of bonds (single, double, etc.) and $M$ the types of atoms (e.g., oxygen, carbon, etc.). Our objective is to train an invertible model $f_\theta$ with parameters $\theta$ that maps $G$ into a latent point $z = f_\theta(G) \in \mathbb{R}^{D=(N \times N \times R)+(N \times M)}$. We describe $f_\theta$ as a normalizing flow composed of multiple invertible functions.

Let $z$ be a latent vector drawn from a known prior distribution $p_z(z)$ (e.g., Gaussian): $z \sim p_z(z)$. After applying a variable transformation, the log probability of a given graph $G$ can be calculated as:

$$\log\left(p_G(G)\right) = \log\left(p_z(z)\right) + \log\left(\left|\det\left(\frac{\partial z}{\partial G}\right)\right|\right), \tag{1}$$

where $\frac{\partial z}{\partial G}$ is the Jacobian of $f_\theta$ at $G$.

In (Madhawa et al., 2019) $f_\theta$ is modeled by two types of invertible non-volume preserving (NVP) mappings (Dinh et al., 2017). The first type of mapping is the one that transforms the adjacency tensor, and the second type is the one that transforms the node attribute $X$.

Let us divide the hidden variable $z$ into two parts $z = [z_X, z_A]$; the former $z_X$ is derived from invertible mappings of $X$ and the latter $z_A$ is derived from invertible mappings of $A$. For the mapping of the feature matrix $X$, the GraphNVP provides a node feature coupling:

$$z_X^{(\ell)}[\ell, :] \leftarrow z_X^{(\ell-1)}[\ell, :] \circ \exp\left(s(z_X^{(\ell-1)}[\ell^-, :], A)\right) + t(z_X^{(\ell-1)}[\ell^-, :], A), \tag{2}$$

where $\ell$ indicates the layer of the coupling, functions $s$ and $t$ stand for scale and translation operations, respectively, and $\circ$ denotes element-wise multiplication. We use $z_X[\ell^-, :]$ to denote a latent representation matrix of $X'$ excluding the $\ell^{\text{th}}$ row (node). The rest of the rows of the feature matrix remains the same as follows.

$$z_X^{(\ell)}[\ell^-, :] \leftarrow z_X^{(\ell-1)}[\ell^-, :]. \tag{3}$$

$s$ and $t$ are modeled by a generic GCN, requiring the adjacency information of nodes, $A$, for better interactions between the nodes.

For the mapping of the adjacency tensor, the GraphNVP provides an adjacency coupling:

$$z_A^{(\ell)}[\ell, :, :] \leftarrow z_A^{(\ell-1)}[\ell, :, :] \circ \exp\left(s(z_A^{(\ell-1)}[\ell^-, :, :])\right) + t(z_A^{(\ell-1)}[\ell^-, :, :]). \tag{4}$$

The rest of the rows remain as they are, as follows:

$$z_A^{(\ell)}[\ell^-, :, :] \leftarrow z_A^{(\ell-1)}[\ell^-, :, :]. \tag{5}$$

For adjacency coupling, we employ simple multi-layer perceptrons (MLPs) for $s$ and $t$.

The abovementioned formulations map only those variables that are related to a node $\ell$ in each $\ell$-th layer (Eqs.(2,4)) , and the remaining nodes $\ell^-$ are kept intact (Eqs.(3,5)); i.e. the partitioning of the variables always occurs in the first axis of tensors. This limits the parameterization of scaling and translation operations, resulting in reduced representation power of the model.

In the original paper, the authors mention: *"masking (switching) ... w.r.t the node axis performs the best. ... We can easily formulate ... the slice indexing based on the non-node axis ... results in dramatically worse performance due to the sparsity of molecular graph."* Here, sparsity can be described in two ways: one is the sparsity of non-carbon atoms in organic chemicals, and the other is the low degrees of atom nodes (because of valency).

## 2.2 INVERTIBLE RESIDUAL BLOCKS

One of the major drawbacks of the partition-based coupling flow is that it covers a fairly limited family of mappings. Instead, the coupling flow offers computational cheap and analytic form of inversions. A series of recent invertible models (Behrmann et al., 2019; Song et al., 2019; Chen et al., 2019) propose a different approach for invertible mappings, called *residual flow* (Kobyzev et al., 2019). They formulate ResNets (He et al., 2016), which have been successful in image recognintion, as invertible mappings. The general idea is described as follows.

Our objective is to develop an invertible residual layer for a vector $z$:

$$z^{(\ell+1)} = z^{(\ell)} + \mathscr{R}\left(z^{(\ell)}\right), \tag{6}$$

where $z^{(\ell)}$ is the representation vector at the $\ell$th layer, and $\mathscr{R}$ is a residual block. If we correctly constrain $\mathscr{R}$, then we can assure the invertibility of the above-mentioned residual layer.

i-ResNet (Behrmann et al., 2019) presents a constraint regarding the Lipschitz constant of $\mathscr{R}$. MintNet (Song et al., 2019) limits the shape of the residual block $\mathscr{R}$ and derives the non-singularity requirements of the Jacobian of the (limited) residual block.

Notably, the (invertible) residual connection (Eq.(6)) does not assume the partition of variables into "intact" and "afine-map" parts. This means that each layer of invertible residual connection updates all the variables at once.

In both the aforementioned papers, local convolutional network architecture (He et al., 2016) of the residual block $\mathscr{R}$ is proposed for image tensor inputs, which can be applied for image generation/reconstructions for experimental validations. For example, in i-ResNet, the residual block is

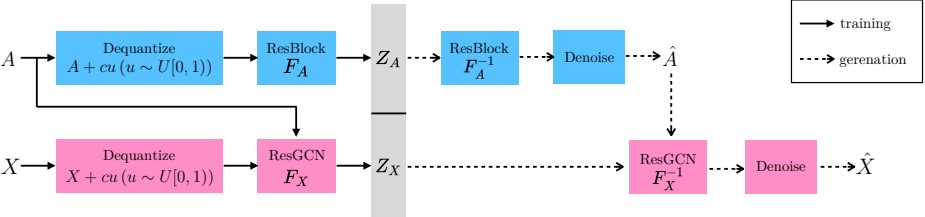

Figure 1: Overall architecture of the proposed generative model. During a forward path (encoding), discrete graph components $A, X$ are inputted for dequantization. We apply the proposed GRFs to the dequantized components and obtain the latent representations, $Z_A, Z_X$. During a backward path (graph generation), we first apply the inversion of the ResBlock to $Z_A$, yielding noise-overlapped $\hat{A}'$. Denoised (recovered) $\hat{A}$ and $Z_X$ are the input arguments for the inverted ResGCN, recovering the noise-overlapped $\hat{X}'$.

defined as:
$$\mathscr{R}(x) = W_3 \circ \phi \circ W_2 \circ \phi \circ W_1(x) , \tag{7}$$
where $\phi$ denotes a contractive nonlinear function such as ReLU and ELU, $W$. are (spatially) local convolutional layers (i.e. aggregating the neighboring pixels). In this case, we put a constraint that the spectral norms of all $W$s are less than unity for the Lipschitz condition.

## 3 INVERTIBLE GRAPH GENERATION MODEL WITH GRAPH RESIDUAL FLOW (GRF)

We observe that the limitations of the GraphNVP cannot be avoided as long we use the partition-based coupling flows for the sparse molecular graph. Therefore we aim to realize a different type of an invertible coupling layer that does not depend on the variable partitioning (for easier inversion and likelihood computation). For this, we propose a new molecular graph generation model based on a more powerful and efficient *Graph Residual Flow* (GRF), which is our proposed invertible flow for graphs.

### 3.1 SETUP

The overall setup is similar to that of the original GraphNVP. We use the notation $G = (A, X)$ to represent a graph $G$ comprising an adjacency tensor $A \in \{0, 1\}^{N \times N \times R}$ and a feature matrix $X \in \{0, 1\}^{N \times M}$. Each tensor is mapped to a latent representation through invertible functions. Let $z_A \in \mathbb{R}^{N \times N \times R}$ be the latent representation of the adjacency tensor, and $p(z_A)$ be its prior. Similarly, let $z_X \in \mathbb{R}^{N \times M}$ be the latent representation of the feature matrix, and $p(z_X)$ be its prior. We assume that both the priors are multivariate normal distributions.

As $A$ and $X$ are originally binary, we cannot directly apply the change-of-variables formula. The widely used (Dinh et al., 2017; Kingma & Dhariwal, 2018; Madhawa et al., 2019) workaround is dequantization: adding noises drawn from a continuous distribution and regarding the tensors as continuous. The dequantized graph denoted as $G' = (A', X')$ is used as the input in Eq. 1:

$$A' = A + cu; \ u \sim U[0, 1)^{N \times N \times R} , \tag{8}$$
$$X' = X + cu; \ u \sim U[0, 1)^{N \times M} , \tag{9}$$

where $0 < c < 1$ is a scaling hyperparameter. We adopted $c = 0.9$ for our experiment.

Note that the original discrete inputs $A$ and $X$ can be recovered by simply applying floor operation on each continuous value in $A'$ and $X'$. Hereafter, all the transformations are performed on dequantized inputs $A'$ and $X'$.

### 3.2 FORWARD MODEL

We can instantly formulate a naive model, and for doing so, we do not take into consideration the graph structure behind $G'$ and regard $A'$ and $X'$ as simple tensors (multi-dimensional arrays).

Namely, an tensor entry $X'[i, m]$ is a neighbor of $X[i', m']$, where $|i' - i| \leq 1$, and $|m' - m| \leq 1$, regardless of the true adjacency of node $i$ and $i'$, and the feature $m$ and $m'$. Similar discussion holds for $A'$.

In such case, we simply apply the invertible residual flow for the tensors $A', X'$. Let $z_A^{(0)} = A'$ and $z_X^{(0)} = X'$.

We formulate the invertible graph generation model based on GRFs. The fundamental idea is to replace the two coupling flows in GraphNVP with the new GRFs. A GRF conmprises two sub-flows: *node feature residual flow* and *adjacency residual flow*.

For the feature matrix, we formulate a *node feature residual flow* for layer $\ell$ as:

$$z_X^{(\ell)} \leftarrow z_X^{(\ell-1)} + \mathscr{R}_X^{(\ell)} \left( z_X^{(\ell-1)}; A \right) , \tag{10}$$

where $\mathscr{R}_X^{(\ell)}$ is a residual block for feature matrix at layer $\ell$. Similar to Eq.(2), we assume the condition of the adjacency tensor $A$ for the coupling.

For the mapping of the adjacency tensor, we have a similar *adjacency residual flow*:

$$z_A^{(\ell)} \leftarrow z_A^{(\ell-1)} + \mathscr{R}_A^{(\ell)} \left( z_A^{(\ell-1)} \right) , \tag{11}$$

where $R_A^{(\ell)}$ is a residual block for adjacency tensor at layer $\ell$.

Note that there are no slice indices of tensors $Z_A$ and $Z_X$ in Eqs.(10, 11). Therefore every entry of the tensors is subject to update in every layer, making a notable contrast with Eqs.(2,4).

## 3.3 Residual Block Choices for GRFs

One of the technical contributions of this paper is the development of residual blocks for GRFs. The convolution architecture of ResNet reminds us of GCNs (e.g. (Kipf & Welling, 2017)), inspiring possible application to graph input data. Therefore, we extend the invertible residual blocks of (Behrmann et al., 2019; Song et al., 2019) to the feature matrix and the adjacency tensor conditioned by the graph structure $G$.

The simplest approach to constructing a residual flow model is by using linear layer as layer $\mathscr{R}$. In such cases, we transform the adjacency matrix and feature matrix to single vectors. However, we must construct a large weight matrix so as not to reduce its dimension. Additionally, naive transformation into vector destroys the local feature of the graphs. To address the aforementioned issues, we propose two types of residual blocks $\mathscr{R}_A$ and $\mathscr{R}_X$ for each of the adjacency matrix and feature matrices.

In this paper, we propose a residual block based on GCNs (e.g. (Kipf & Welling, 2017; Wu et al., 2019) for graph-structured data. We focus on modeling the residual block for the node feature matrix.

The original GCN (Kipf & Welling, 2017) perform the convolution on graphs using the adjacency information of the graph (plus a weight matrix). One layer of the GCN performs the following update of graph node representation $z_X$:

$$z_{X,r}^\ell \leftarrow \phi \left( \tilde{D}_r^{-\frac{1}{2}} \tilde{A}_r \tilde{D}_r^{-\frac{1}{2}} z_X^{\ell-1} W_r \right) , \quad where \tilde{D}_r = D_r + I , \tilde{A}_r = A_r + I , \tag{12}$$

where $A_r \in \mathbb{R}^{N \times N}$ is an adjacency matrix of the graph defined by the relation type $r$, $D \in \mathbb{R}^{N \times N}$ is a degree matrix: $D = \text{diag}(\sum_j A_r ii)_i$[1], $W_r$ is the learnable weight matrix for the relation $r$, and $\phi$ is a nonlinear function. In a nutshell, Eq.(12) updates the each node representation in $z_X$ by the weighed sum of neighbor nodes defined by the adjacency tensor $A_r$.

For our residual blocks on the graph, we replace the convolution filter $W$ in Eq.(7) with Eq.(12) to define the neighbors on a graph:

$$\mathscr{R}_X \left( z_X; r \right) = \phi \left( \text{vec} \left( \tilde{D}_r^{-1/2} \tilde{A}_r \tilde{D}_r^{-1/2} X W_r \right) \right) , \tag{13}$$

$$\mathscr{R}_X \left( z_X \right) = \text{sum}_r \mathscr{R}_X \left( z_{X:r} \right) , \tag{14}$$

---

[1]If an entry of $\tilde{D}_r$ is 0, then we assume the corresponding entry of $\tilde{D}_r^{-\frac{1}{2}}$ is also 0.

where vec is a vectorize operator, and mat is a matrizie operaton. e $z_X$ appropriately. For $\mathscr{R}_X$ defined in this way, the following theorem holds.

**Theorem 1.** $\mathrm{Lip}(\phi) \le L, \|W\|_{\mathrm{op}} < \dfrac{1}{L} \Rightarrow \mathrm{Lip}(\mathscr{R}_X) < 1$.

Here, $\mathrm{Lip}(\cdot)$ is a Lipschitz-constant of a certain function. The proof of this theorem is provided in appendix.

The Lipschitz constraint not only enables inverse operation (see Section 3.4) but also facilitates the computation of the log-determinant of Jacobian matrix in Eq. (1) as performed in (Behrmann et al., 2019). In other words, the log-determinant of Jacobian matrix can be approximated to the matrix trace (Withers & Nadarajah, 2010), and the trace can be computed through power series iterations and stochastic approximation (Hutchinson's trick) (Hall, 2015; Hutchinson, 1990).

## 3.4 BACKWARD MODEL OR GRAPH GENERATION

As our model is invertible, the graph generation process is as depicted in Fig.1. The adjacency tensors and the feature tensors can be simultaneously calculated during training, because their calculations are independent of each other. However, we must note that during generation, a valid adjacency tensor is required for the inverse computation of ResGCN. For this reason, we execute the following 2-step generation: first, we generate the adjacency tensor and subsequently generate the atomic feature tensor. The abovementioned generation process is shown in the right half of Fig.1. The experiment section shows that this two-step generation process can efficiently generate chemically valid molecular graphs.

**1st step:** We sample $z = \mathrm{concat}(z_A, z_X)$ from prior $p_z$ and split the sampled $z$ into two, one of which is for $z_A$ and the other is for $z_X$. Next, we compute the inverse of $z_A$ w.r.t Residual Block by fixed-point-iteration. Consequently, we obtain a probabilistic adjacency tensor $\hat{A}'$. Finally, we construct a discrete adjacency tensor $\hat{A} \in \{0,1\}^{N \times N \times R}$ from $\hat{A}'$ by taking node-wise and edge-wise argmax operation.

**2nd step:** We consider the discrete matrix $\hat{A}$ obtained above as a fixed parameter and calculate the inverse image of $z_X$ for ResGCN using fixed-point iteration. In this way, we obtain the probabilistic adjacency tensor $\hat{X}'$. Next, we construct a discrete feature matrix $\hat{X} \in \{0,1\}^{N \times M}$ by taking node wise argmax operation. Finally, we construct the molecule from the obtained adjacency tensor and feature matrix.

### 3.4.1 INVERSION ALGORITHM: FIXED POINT ITERATION

For the residual layer $f(x)(= x + \mathscr{R}(x))$, it is generally not feasible to compute the inverse image analytically. However, we have configured the layer to satisfy $\mathrm{Lip}(\mathscr{R})$ as described above. As was done in the i-ResNet (Behrmann et al., 2019), the inverse image of $f(x)$ can be computed using a fixed-point iteration of Algorithm 1 in appendix. From the Banach fixed-point theorem, this iterative method converges exponentially.

### 3.4.2 CONDITION FOR GUARANTEED INVERSION

From theorem 1, the upper bound of $\mathrm{Lip}(\mathscr{R}_X)$ is determined by $\mathrm{Lip}(\phi)$ and $\|W\|_{\mathrm{op}}$. In this work, we selected the exponential linear unit (ELU) as function $\phi$. ELU is a nonlinear function, which satisfies the differentiability condition. By definition, $\mathrm{Lip}(\mathrm{ELU}) = 1$. For W, the constraints can be satisfied by using spectral normalization (Miyato et al., 2018). The layer $\mathscr{R}_X$ configured in this manner holds $\mathrm{Lip}(\mathscr{R}_X) < 1$. In other words, this layer is the contraction map. Here, the input can be obtained by fixed point iteration.

## 4 EXPERIMENTS

### 4.1 PROCEDURE

For our experiments, we use two datasets of molecules, QM9 (Ramakrishnan et al., 2014) and ZINC-250k (Irwin et al., 2012). The QM9 dataset contains 134,000 molecules with four atom types, and ZINC-250k is a subset of the ZINC-250k database that contains 250,000 drug-like molecules with nine atom types. The maximum number of heavy atoms in a molecule is nine for the QM9 and 38 for the ZINC-250k. As a standard preprocessing, molecules are first kekulized and the hydrogen atoms are subsequently removed from these molecules. The resulting molecules contain only single, double, or triple bonds.

We represent each molecule as an adjacency tensor $A \in \{0,1\}^{N \times N \times R}$ and a one-hot feature matrix $X \in \{0,1\}^{N \times M}$. $N$ denotes the maximum number of atoms a molecule in each dataset can have. If a molecule has less than $N$ atoms, it is padded by adding virtual nodes to keep the dimensions of $A$ and $X$ identical. As the adjacency tensors of molecular graphs are sparse, we add virtual bonds, referred to as "no bond," between the atoms that do not have a bond.

Thus, an adjacency tensor conmprises $R=4$ adjacency matrices stacked together. Each adjacency matrix corresponds to the existence of a certain type of bond (single, double, triple, and virtual bonds) between the atoms. The feature matrix represents the type of each atom (e.g., oxygen, fluorine, etc.). As described in Section 3.3, $X$ and $A$ are dequantized to $X'$ and $A'$.

We use a standard Gaussian distribution $\mathcal{N}(\mathbf{0}, \boldsymbol{I})$ as a prior distribution $p_z(z)$. The objective function (1) is maximized by the Adam optimizer (Kingma & Ba, 2015). The hyperparameters are chosen by optuna (Akiba et al., 2019) for QM9 and ZINC-250k. Please find the appendix for the selected hyperparameter values. To reduce the model size, we adopt node-wise weight sharing for QM9 and low-rank approximation and multi-scale architecture proposed in (Dinh et al., 2017) for ZINC-250k.

### 4.2 INVERTIBILITY CHECK

We first examine the reconstruction performance of GRF against the number of fixed-point iterations by encoding and decoding 1,000 molecules sampled from QM9 and ZINC-250k. According to Figure 2b, the L2 reconstruction error converges around $10^{-4}$ after 30 fixed point iterations. The reconstructed molecules are the same as the original molecule after convergence.

### 4.3 NUMERICAL EVALUATION

Following (Kingma & Dhariwal, 2018; Madhawa et al., 2019), we sample 1,000 latent vectors from a temperature-truncated normal distribution $p_z(z; T_X, T_A)$ and transform them into molecular graphs by inverse operations. Different temperatures are selected for $X$ and $A$ because they are handled separately in our model. We compare the performance of the proposed model with those of the baseline models using the following metrics. **Validity (V)** is the ratio of the chemically valid molecules to the generated graphs. **Novelty (N)** is the ratio of the molecules that are not included in the training set to the generated valid molecules. **Uniqueness (U)** is the ratio of the unique molecules to the generated valid molecules. **Reconstruction accuracy (R)** is the ratio of the molecules that are reconstructed perfectly by the model. This metric is not defined for GANs as they do not have encoders.

We choose GraphNVP (Madhawa et al., 2019), Junction Tree VAE (JT-VAE) (Jin et al., 2018), Regularizing-VAE (RVAE) (Ma et al., 2018) as state-of-the-art baseline models. Also, we choose two additional VAE models as baseline models; grammar VAE(GVAE) (Kusner et al., 2017) and character VAE (CVAE) (Gómez-Bombarelli et al., 2018), which learn SMILES(string) representations of molecules.

We present the numerical evaluation results of QM9 and ZINC-250K datasets on the Table 1 (QM9) and the Table 2 (ZINC-250K), respectively. As expected, GRF achieves 100% reconstruction rate, which is enabled by the ResNet architecture with spectral normalization and fixed-point iterations. This has never been achieved by any other VAE-based baseline that imposes stochastic behavior in the bottleneck layers. Also, this is achieved without incorporating the chemical knowledge, which is done in some baselines (e.g., valency checks for chemical graphs in RVAE and GVAE, subgraph

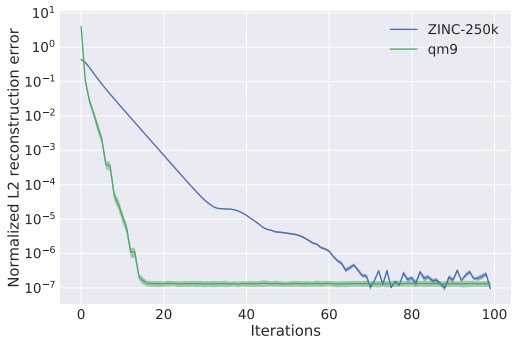
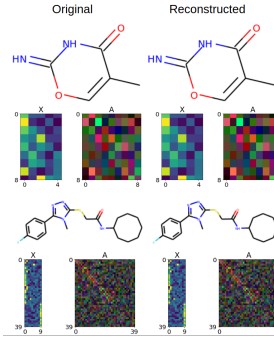

(a) Reconstruction error.

(b) Original and reconstructed molecules.

Figure 2: (a) L2 reconstruction error of GRF against the number of fixed-point iterations. L2 errors are measured between dequantized $A'$, $X'$ and reconstructed $\hat{A}'$, $\hat{X}'$, normalized by the number of entries. We observe that the error decays exponentially with the number of iterations in both datasets. (b) The original and reconstructed molecules sampled from QM9 and ZINC-250k with 100 fixed-point iterations. The color maps depict the values of the dequantized feature and adjacency matrices. Feature matrices have single channel while adjacency matrices have three channels excluding the virtual bond channel and are visualized as RGB channels. Because the values are to be quantized by argmax function over each node and the noise scaling hyperparameter is set as $c = 0.9$ , if at any pixel the value difference is less than 0.1, the molecule is accurately reconstructed.

vocabulary in JT-VAE). This is preferable because additional validity checks are computationally demanding, and the prepared subgraph vocabulary limits the extrapolation capacity of the generative model. As our model does not incorporate domain-specific procedures, it can be easily extended to general graph structures.

It is remarkable that our GRF-based generative model achieves good generation performance scores comparable to GraphNVP, with *much fewer trainable parameters in order of magnitude*. These results indicate the efficient construction of our GRF in terms of parametrization, as well as powerfulness and flexibility of the residual connections, compared to the coupling flows based on simple affine transformations. Therefore, our goal of proposing a novel and strong invertible flow for molecular graph generation is successfully achieved by the development of the GRF. We will discuss the number of parameters of GRF using Big-O notation in Section 4.4.

The experiments also reveal a limitation of the current formulation of the GRF. One notable limitation is the lower uniqueness compared to the GraphNVP. We found that the generated molecules contain many straight-chain molecules compared to those of GraphNVP, by examining the generated molecules manually. We attribute this phenomenon to the difficulty of generating realistic molecules without explicit chemical knowledge or autoregressive constraints. We are planning to tackle this issue as one of the future works.

## 4.4 EFFICIENCY IN TERMS OF MODEL SIZE

As we observe in the previous section, our GRF-based generative models are *compact* and *memory-efficient* in terms of the number of trainable parameters, compared to the existing GraphNVP flow model. In this section we discuss this issue in a more formal manner.

Let $L$ be the number of layers, $R$ be the number of the bond types, $M$ be the number of atom types. For GraphNVP, We need $O\left(LN^4R^2\right)$ and $O\left(LN^2M^2R^2\right)$ parameters to construct adjacency coupling layers and atom coupling layers, respectively. From the above, we need $O\left(LN^2R^2(N^2 + M^2)\right)$ parameters to construct whole GraphNVP. By contrast, our model only requires $O\left(LR^2N^2\right)$ and $O\left(LR^2M^2\right)$ parameters for res-GraphLinear and res-GCN, respectively. Therefore, whole GRF model requires $O\left(LR^2(N^2 + M^2)\right)$ parameters. In most cases of molecular graph generation settings, $R \leq 5$ and $N$ is dominant.

Table 1: Performance of generative models with respect to quality metrics and numbers of their parameters for QM9 dataset. Results of GraphNVP are recomputed following the hyperparameter setting in the original paper. Other baseline scores are borrowed from the original papers. Scores of GRF are averages over 5 runs. Standard deviations are presented below the averaged scores. We use $T_X = 0.65$ and $T_A = 0.69$ for QM9.

| Method | % V | % N | % U | % R | # Params |
|---|---|---|---|---|---|
| **GRF** | 84.5 ($\pm$ 0.70) | 58.6 ($\pm$ 0.82) | 66.0 ($\pm$ 1.15) | 100.0 | 56,120 |
| GraphNVP | 90.1 | 54.0 | 97.3 | 100.0 | 6,145,831 |
| RVAE | 96.6 | 97.5 | - | 61.8 | - |
| GVAE | 60.2 | 80.9 | 9.3 | 96.0 | - |
| CVAE | 10.3 | 90.0 | 67.5 | 3.6 | - |

Table 2: Performance of generative models with respect to quality metrics and numbers of their parameters for ZINC-250K dataset. Results of GraphNVP and JT-VAE are recomputed following the hyperparameter setting in the original paper. Other baseline scores are borrowed from the original papers. Scores of GRF are averages over 5 runs. Standard deviations are presented below the averaged scores. We use $T_X = 0.15$ and $T_A = 0.17$ for ZINC-250k.

| Method | % V | % N | % U | % R | # Params |
|---|---|---|---|---|---|
| **GRF** | 73.4 ($\pm$ 0.62) | 100.0 ($\pm$ 0.0) | 53.7 ($\pm$ 2.13) | 100.0 | 3,234,552 |
| GraphNVP | 77.3 | 100.0 | 94.8 | 100.0 | 245,792,665 |
| JT-VAE | 99.8 | 100.0 | 100.0 | 76.7 | - |
| RVAE | 34.9 | 100.0 | - | 54.7 | - |
| GVAE | 7.2 | 100.0 | 9.0 | 53.7 | - |
| CVAE | 0.7 | 100.0 | 67.5 | 44.6 | - |

Our GRF for ZINC-250k uses linear layers to handle adjacency matrices, but the number of the parameters is substantially reduced by low-rank approximation (introduced in Sec. 4.1). Let $r$ be the approximated rank of each linear layer, and the whole GRF requires only $O\left(LR^2(N^2r + M^2)\right)$ parameters. Notably, GraphLinear is equal to low-rank approximation when $r = 1$.

Our model's efficiency in model size is much more important when generating large molecules. Suppose we want to generate molecule with $N = 100$ heavy atoms with batch size of 64. Estimating from the memory usage of GRF for ZINC-250k ($N = 40$), GRF will consume 21 GB if $r = 100$ and GraphNVP will consume as large as 2100 GB. Since one example of the GPUs currently used (e.g., NVIDIA Tesla V100) is equipped with $16 - 32$ GB memory, GraphNVP cannot process a batch on a single GPU or batch normalization becomes unstable with small batch. On the other hand, our model will scale to larger graphs due to the reduced parameters.

## 4.5 Smoothness of the Learned Latent Space

As a final experiment, we present the visualization of the learned latent space of $Z$. First we randomly choose 100 molecules from the training set, and subsequently encode them into the latent representation using the trained model. We compute the first and the second principal components of the latent space by principal component analysis (PCA), and project the encoded molecules onto the plane spanned by these two principal component vectors. Then we choose another random molecule, $x_o$, encode it and project it onto the aforementioned principal plane. Finally we decode the latent points on the principal plane, distributed in a grid-mesh pattern centered at the projection of $x_o$, and visualize them in Fig. 3. Figure 3 indicates that the learnt latent spaces from both QM9 (panel (a)) and ZINC-250k datasets (panel (b)) are smooth where the molecules gradually change along the two axes.

The visualized smoothness appears to be similar to that of the VAE-based models but differs in that our GRF is a bijective function: the data points and the latent points correspond to each other in a one-to-one manner. In contrast, to generate the data points with VAE-based methods, it is required to decode the same latent point several times and select the most common molecule. Our model is

efficient because it can generate the data point in one-shot. Additionally, smooth latent space and bijectivity are crucial to the actual use case. Our model enables molecular graph generation through querying: encode a molecule with the desired attributes and decode the perturbed latents to obtain the drug candidates with similar attributes.

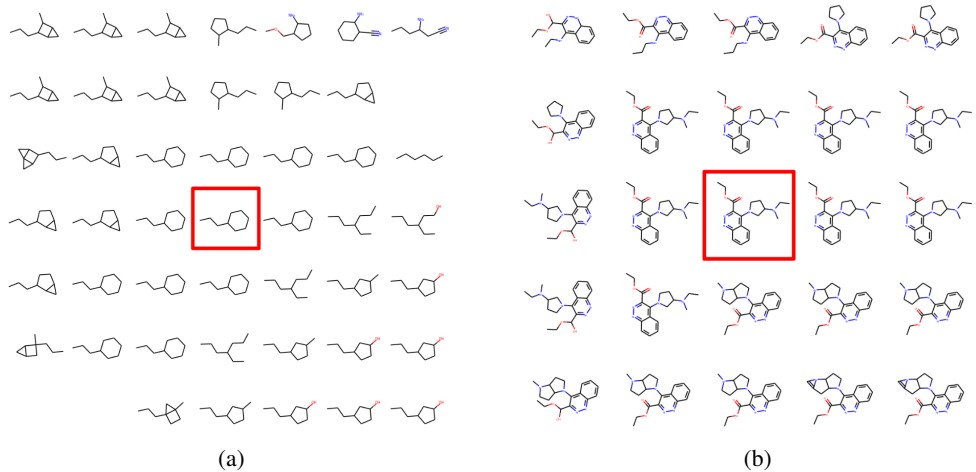

(a)                                   (b)

Figure 3: Visualization of the learned latent spaces along the first and the second principal components. The empty space in the grid indicates that an invalid molecule is generated.

## 5 CONCLUSION

In this paper, we proposed a Graph Residual Flow, which is an invertible residual flow for molecular graph generations. Our model exploits the expressive power of ResNet architecture. The invertibility of our model is guaranteed only by a slight modification, i.e. by the addition of spectral normalization to each layer. Owing to the aforementioned feature, our model can generate valid molecules both in QM9 and ZINC-250k datasets. The reconstruction accuracy is inherently 100%, and our model is more efficient in terms of model size as compared to GraphNVP, a previous flow model for graphs. In addition, the learned latent space of GRF is sufficiently smooth to enable the generation of molecules similar to a query molecule with known chemical properties.

Future works may include the creation of adjacency residual layers invariant for node permutation, and property optimization with GRF.

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

## A    PROOF OF THEOREM

**Lemma 1.** $^\forall A \in \mathbb{R}^{N \times N}, ^\forall X (= [x_1, \ldots, x_N]) \in \mathbb{R}^{N \times d}, s.t. \|AX\|_F \le \|A\|_{\mathrm{op}} \|X\|_F$.

*Proof.*

$$\|AX\|_F^2 = \sum_{i=1}^{N} \|Ax_i\|_2^2$$

$$\le \|A\|_{\mathrm{op}}^2 \sum_{i=1}^{N} \|x_i\|_2^2$$

$$= \|A\|_{\mathrm{op}}^2 \|X\|_F^2 .$$

$$\therefore \|AX\|_F \le \|A\|_{\mathrm{op}} \|X\|_F$$

$\square$

**Lemma 2.** $\|P\|_{\mathrm{op}} = \left\| \tilde{D}^{-1/2} \tilde{A} \tilde{D}^{-1/2} \right\|_{\mathrm{op}} \le 1$.

*Proof.* Augmented Normalized Laplacian $\tilde{L}$ is defined as $\tilde{L} = I - \tilde{D}^{-1/2} \tilde{A} \tilde{D}^{-1/2} = I - P$. Like the normal graph Laplacian, an $i$-th eigenvalue $\tilde{\mu}_i$ of $\tilde{L}$ holds $0 \le \tilde{\mu}_i \le 2$ (Oono & Suzuki, 2019). Here, for the eigenvector $v_i$ corresponding to $\lambda_i$, which is the i-th eigenvalue of $P$:

$$Pv_i = \lambda_i v_i$$

$$v_i - \tilde{L} v_i = \lambda_i v_i$$

$$\tilde{L} v_i = (1 - \lambda_i) v_i$$

$$\therefore \lambda_i = 1 - \tilde{\mu}_i.$$

As $0 \le \tilde{\mu}_i \le 2$, $-1 \le \lambda_i \le 1$ i.e. $|\lambda_i| \le 1$ follows. Here, operation norm $\|P\|_{\mathrm{op}}$ is bounded maximum singular value $\sigma(P)$. As $P$ is a symmetric matrix from its construction, the maximum singular value $\sigma(P)$ is equal to the absolute eigenvalue $|\lambda_{\max}|$ with the largest absolute value. From these conditions, $\|P\|_{\mathrm{op}} \le \sigma(P) = |\lambda_{\max}| \le 1$.

$\square$

**Theorem 1.** $\mathrm{Lip}(\phi) \le L, \|W\|_{\mathrm{op}} < \dfrac{1}{L} \Rightarrow \mathrm{Lip}(\mathscr{R}_X) < 1$.

*Proof.*

$$\begin{aligned}
\|\mathscr{R}_X(x) - \mathscr{R}_X(y)\|_2 &= \|\phi(\mathrm{vec}(PXW)) - \phi(\mathrm{vec}(PYW))\|_2 \\
&\le L \|\mathrm{vec}(PXW) - \mathrm{vec}(PYW)\|_2 && (\because \mathrm{Lip}(\phi) \le L) \\
&= L \|PXW - PYW\|_F \\
&= L \|P(X - Y)W\|_F \\
&\le L \|P\|_{\mathrm{op}} \|(X - Y)W\|_F && (\because \text{Lemma 2.}) \\
&\le L \|(X - Y)W\|_F && (\because \text{Lemma 1.}) \\
&\le L \|W\|_{\mathrm{op}} \|X - Y\|_F \\
&< L \cdot \frac{1}{L} \|X - Y\|_F \\
&\le \|X - Y\|_F \\
&= \|\mathrm{vec}(X) - \mathrm{vec}(Y)\|_2 \\
&= \|x - y\|_2 .
\end{aligned}$$

$$\therefore \mathrm{Lip}(\mathscr{R}_X) < 1.$$

$\square$

---

**Algorithm 1** Inverse of Residual-layer via fixed-point iteration.

---

**Input:** output from residual layer $y$, contractive residual block $\mathscr{R}$, number of iterations $n$
**Output:** inverse of $y$ w.r.t $\mathscr{R}$
$x_0 \leftarrow y$
**for** $i = 0, \dots, n$ **do**
   $x_{i+1} \leftarrow y - \mathscr{R}(x_i)$
**end for**
return $x_n$

---

## B    MODEL HYPERPARAMETERS

We use a single-scale architecture for QM9 dataset, while we use multi-scale architecture (Dinh et al., 2017) for ZINC-250k dataset to scale to 38 heavy atoms. Other hyperparameters are shown in Table 3. We find the factor of spectral normalization 0.9 is enough for numerical invertibility.

Table 3: Model hyperparameters for QM9 and ZINC250k. BS and LR stand for batch size and learning rate, respectively.

| Dataset | GCN blocks | GCN layers | MLP blocks | MLP layers | BS | LR | Epochs |
|---|---|---|---|---|---|---|---|
| QM9 | 1 | 1 | 32 | 25 | 2048 | 1e-3 | 70 |
| ZINC-250k | 3 | 3 | 3 | 3 | 256 | 1e-4 | 70 |

## C    ALGORITHMS

We show a procedure to calculate inverse of $y$ with reference to $\mathscr{R}$ in algorithm 1. In experiments, we chose 100 as number of iteration $n$.

