# OpenReview forum: "Graph Residual Flow for Molecular Graph Generation"
_ICLR.cc/2020/Conference — Reject_

### Official Review · AnonReviewer3 · 2019-10-21
**Official Blind Review #3**

**Rating:** 3

**Review:**

Summary:

The paper proposes a flow-based model for generating molecules, called Graph Residual Flow (GRF). GRF is a version of iResNet that is tailored to molecule generation. The experiments show that GRF achieves similar performance to GraphNVP (a previous flow-based model for molecule generation), but has about 100 times fewer parameters.

Decision:

This is interesting work, but in my opinion the current version of the paper needs significant improvement. For this reason, my recommendation is weak reject.

Contribution & originality:

I think the application of flow models to the generation of molecules is an interesting research direction. Since molecules are structured objects that can be represented as graphs of connected atoms, care has to be taken to design suitable generative models for them, and this paper is a step in this direction.

The proposed model (GRF) is similar to iResNet in most aspects. The main differences are:
- The linear layers in iResNet are modified to take into account the connectivity of the molecule.
- Separate networks are used to generate the adjacency tensor and the feature matrix. The adjacency tensor is generated first, and then the feature-matrix network is conditioned on the generated adjacency matrix.
Other than the above, GRF is a straightforward application of iResNet in molecule generation.

Technical & writing quality:

The experiments are interesting, and GRF is shown to perform similarly to GraphNVP while using 100 times fewer parameters. The comparison between GRF and GraphNVP is in my opinion the most interesting part of the paper. The experiments that show GRF can reconstruct molecules exactly and that generated molecules are a smooth function of the latent variables are less interesting in my opinion; exact invertibility and smoothness are inherent properties of flow-based models that exist by design, so there isn't anything surprising about them.

The main issue with the paper is that it's poorly written. In my opinion, the writing quality of the paper needs significant improvement for the paper to qualify for publication. The model is explained very poorly; if I weren't already familiar with iResNet I don't think I would be able to understand the model and be able to reproduce the results by reading the paper. Moreover, the use of English throughout the paper is poor: there are a lot of grammatical mistakes (such as incorrect usage of articles or lack of subject-verb agreement), and awkward expressions. In the following I will do my best to make concrete suggestions for improvement, but I would strongly encourage the authors to revise the paper thoroughly and have it checked for grammar and writing quality.

Suggestions for improvement:

"The decoders of these graph generation models generates" --> generate

"The state-of-the-art VAE-based models [...] have good generation performance but their decoding scheme is highly complicated"
In what way is it complicated? Please provide more details.

"invertible flow-based statistical models [...] does" --> do

"[flow-based models don't] require training for their decoders because the decoders are simply the inverse mapping of the
encoders"
This is an odd and unusual description of flow-based models. The way I see it, flow-based models aren't autoencoders, but functions that reparameterize data (here molecules) in terms of simpler random variables (here Gaussian noise). Saying that flow-based models don't require training for their decoders makes little sense.

"Liu et al. (2019) proposes" -->  propose

"GraphNVP has a serious drawback when applied for sparse graphs" --> applied to sparse graphs

"only one affine mapping is executed for each attribute of the latent representation"
There are also coupling layers that are non-affine and therefore more flexible. For example:
- Neural Autoregressive Flows, https://arxiv.org/abs/1804.00779
- Sum-Of-Squares Polynomial Flow, https://arxiv.org/abs/1905.02325
- Neural Spline Flows, https://arxiv.org/abs/1906.04032

"show that a generative model based on the GRF has much fewer trainable parameters compared to the GraphNVP, while still maintaining a comparable generation performance"
If GRF can achieve a performance similar to GraphNVP with fewer parameters, shouldn't it be able to achieve much better performance with more parameters? If this is true, I would have expected to see an experiment were GRF uses more parameters and as a result outperforms GraphNVP. If GRF doesn't outperform GraphNVP even with more parameters, then I would expect to see a discussion explaining why.

it would be good to explain fully the format of the data as early as in section 2.1. For example, it wasn't clear to me until much later how a flow-based model can be applied to discrete data, and whether the rows of the adjacency tensor and feature matrix are one-hot or not. Currently, the format of the data is partially explained three times (in section 2.1, section 3.1 and section 4.1) and only in section 4.1 things become clear for the reader.

Beginning of page 3: the dequantized matrix X' is used before having been defined. This is an example of why the format of the data should be fully explained early on.

"The abovementioned formulations map only those variables [..] (Eqs.(2,5), and the remaining nodes [..] (Eqs.(3,5)"
The first reference should be Eqs. (2,4). Also, the parentheses should close.

"ResNets [...], the golden standard for image recognition"
The expression is "gold standard" not "golden standard". Also, it's too strong a statement and rather subjective to say that ResNets are the gold standard for image recognition; better say that ResNets have been successful in image recognition, which is an accurate and objective statement.

Section 2.2 introduces iResNets, but it doesn't explain clearly that making the residual block contractive is a sufficient condition for invertibility. This is a crucial element for understanding iResNet that is used later on in the paper, so it should be clearly explained early on.

"i-ResNet [...] presents a constraint regarding the Lipschitz constant"
Too vague, and it doesn't explain what the constraint is.

"MintNet [...] derives the non-singularity requirements of the Jacobian of the (limited) residual block"
Also vague and not informative.

Beginning of section 3: GraphMVP --> GraphNVP

"we cannot directly apply the change-of-variables formula directly"
Uses "directly" twice.

"we do not take it consideration" --> we do not take into consideration

"an tensor entry X'[i, m] is a neighborhood" --> a tensor entry X'[i, m] is a neighbor

"Similar discussion holds for A'"
It's unclear to me what exactly holds for A', please be more specific. The previous statement mentioned neighbours, how does this apply to A'?

In the text after eq. (10), R_X uses a different font.

"is subject to update in every layers" --> in every layer

Section 3.3. is particularly poorly written. Given that this is one of the main contributions of the paper, it is really important that this section is written well and clearly. Currently, the section contains two one-sentence paragraphs that are clearly out of place. Also, eq. (12) makes little sense. In particular:
- Isn't A a tensor of shape NxNxR? How did it become a matrix?
- In what sense is X a matrix representation of z_x? How is X computed? Shouldn't X be of shape NxM? Also, please use a different symbol other than X, as X is already used for the feature matrix.
- What is D and how is it defined?
- Is vec() the vectorization operator? Does that mean that the input and output to the residual block has been flattened? This has not been explained or mentioned anywhere.
- Is W a fully-connected linear layer or a convolution as in iResNet?
- Where did the term D^{-0.5} A D^{-0.5} come from? How is it motivated / derived? Why does it make sense to use it? I understand that it's related to the graph Laplacian, but more information is needed for readers who are not familiar with graphs.

"is a Lipschitz-constant of for a certain function" --> of a certain function

The explanation of eq. (13) is very poor, and I don't think a reader who is not already familiar with iResNet can follow it. Also, how was the infinite sum approximated? Did you use truncation or the Russian-roulette estimator?

Section 3.4 introduces the term "atomic feature tensors" which is not defined. I presume it's the same as the feature matrix. Please be consistent with terminology throughout the paper.

"adjacent tensor" --> adjacency tensor

"we have configure" --> configured

Section 3.4.2 finally explains the invertibility condition. Clearly, this is an important element of the algorithm, and should be explained fully and clearly early on. By this point, the invertibility condition should be clear, and no further explanation should be needed.

"we selecte" --> selected

"satisfies the differentiability" --> "satisfies the differentiability condition", or even better "is differentiable"

What does "kekulized" mean?

"conmprises" --> comprises

"we adopt node-wise weight sharing for QM9 and low-rank approximation and multi-scale architecture"
These are important aspects of the architecture that are nowhere else mentioned, and should be explained more thoroughly.

"p_z(T_X, T_A; z)"
Shouldn't this be p_z(z; T_X, T_A)?

What does "unique molecules" mean? Are they the molecules that don't exist in the data? How do they differ from "novel molecules"?

It would be good to provide additional results. In particular:
- It would be good to report training/validation/test log likelihood, or learning curves, that compare between GRF and GraphNVP.
- It would be good to show how the metrics V, N, U, R in tables 1 and 2 vary with respect to the temperatures (for example, show curves of the four metrics as temperatures vary). Currently, results are reported for only one setting of temperatures, which doesn't show how sensitive the performance is to the temperatures.

The temperatures for ZINC-250k are really low (0.15 and 0.17), which to me is an indication that the flow model may have fitted the data quite badly.

How does GRF compare to GraphNVP in terms of generation time? How slow is it to generate from GRF given that it takes 30 iterations for each residual block?

What is GraphLinear? Maybe give a citation?

"Since one of the most GPUs currently used"
This phrase doesn't make much sense.

"principle components analysis" --> principal components analysis
"Principle" has a different meaning from "principal".

Figure 3: Wouldn't it be more informative to centre the visualizations at the mean and use the two principal axes, rather than random mean and axes?

**Experience Assessment:**

I have published one or two papers in this area.

**Review Assessment: Checking Correctness Of Derivations And Theory:**

I carefully checked the derivations and theory.

**Review Assessment: Checking Correctness Of Experiments:**

I carefully checked the experiments.

**Review Assessment: Thoroughness In Paper Reading:**

I read the paper thoroughly.

---

> ### Author Response · Authors · 2019-11-13
> **Response to Reviewer #3**
>
> We thank the reviewer for their comments and questions. In particular, thank you for precious suggestions on our writing. We revised our paper following your suggestions.
> Except for these points, we address them in order.
>
> > Other than the above, GRF is a straightforward application of iResNet in molecule generation.
>
> Unlike images, graphs are discrete and do not have grid-like structures. Therefore, i-ResNet cannot be applied to graphs in a  straightforward way. To address these critical difference, we dequantise adjacency tensors and feature tensors and derive a condition where GRF is invertible reflecting topological information of graphs. From Lemma 1 and 2, the upper bound of the Lipschitz constant of graph residual layers can be decided only on the Lipschitz constant of the activation function (Theorem 1).
>
> >  Exact invertibility and smoothness are inherent properties of flow-based models that exist by design, so there isn't anything surprising about them.
>
> The invertibility and smoothness of GRF is guaranteed in theory, but it is not in reality because fixed point iteration is truncated and some approximations are used in the computation. We conducted those experiments in order to demonstrate that the GRF can be appropriately trained with the derived condition for invertibility.
>
> > If GRF can achieve a performance similar to GraphNVP with fewer parameters, shouldn't it be able to achieve much better performance with more parameters? If GRF doesn't outperform GraphNVP even with more parameters, then I would expect to see a discussion explaining why.
>
> We observed that increased parameters resulted in instability during training (especially in ZINC-250k). Specifically, with increased parameters, the learnt prior p(z) approached to the delta function rather than standard normal distribution. We assume that this is because the model was overfitting to maximize the log p(z) term of the objective function. Some regularization can alleviate this problem (e.g., adding KL divergence penalty), but we leave it for future work.
>
> >  What does "kekulized" mean?
>
> “kekulize” is to replace aromatic bonds in molecules with single/double bonds.
>
> > "We adopt node-wise weight sharing for QM9 and low-rank approximation and multi-scale architecture for ZINC-250k." These are important aspects of the architecture that are nowhere else mentioned, and should be explained more thoroughly.
>
> “Node-wise weight sharing” and “low-rank approximation” is introduced to reduce parameters for fully-connected layers dealing with adjacency tensors. Namely, the weight matrix is expressed as:
> $$W \in  \mathbb{R}^{N \times M} = AB$$
> where $A \in \mathbb{R}^{N \times r}$ and $B \in \mathbb{R}^{r \times M}$. When $r=1$, it is the same as node-wise weight sharing.
> For multi-scale architecture, we refer NICE paper (Dinh et al., ICLR, 2015).
>
>
> >  It would be good to show how the metrics V, N, U, R in tables 1 and 2 vary with respect to the temperatures (for example, show curves of the four metrics as temperatures vary).
>
> We have uploaded the figures here.
> https://drive.google.com/drive/folders/1XZEGyrIulVHZVXGzjaJW_M2Tbzcv7oGG?usp=sharing
> These figures show how the metrics change against the temperature. Note that the temperature is different from that in 4.3. In ZINC-250k, the novelty line overlaps the uniqueness line.
>
> > The temperatures for ZINC-250k are really low (0.15 and 0.17), which to me is an indication that the flow model may have fitted the data quite badly.
>
> This is also caused by the the learnt prior p(z) approaching to the delta function discussed above. Some regularization can alleviate this problem (e.g., adding KL divergence penalty), but we leave it for future work.
>
> > How does GRF compare to GraphNVP in terms of generation time? How slow is it to generate from GRF given that it takes 30 iterations for each residual block?
>
> In the experiments GRF takes 100 iterations for each block and takes around 20 seconds to generate 1,000 graphs (GraphNVP takes around 5 seconds). Nevertheless, it is much faster than autoregression (e.g., JT-VAE takes around 200 seconds).
>
> > What is GraphLinear? Maybe give a citation?
>
> In the revised version, we call it “graph convolution” and please refer Neural Fingerprint (Duvenaud et al., NIPS, 2015).
>
> > Figure 3: Wouldn't it be more informative to centre the visualizations at the mean and use the two principal axes, rather than random mean and axes?
>
> We apologize for the mistake in the caption. As described in 4.5, we use a random molecule as the centered one and two principal axes (fixed in the revised version).

---

### Official Review · AnonReviewer2 · 2019-10-24
**Official Blind Review #2**

**Rating:** 3

**Review:**

The paper introduces an invertible deep generative model architecture for modeling molecular graphs. The model is based on graph residual flows (GRF), which is a graph variant of normalizing flows. The GRF model is a refinement of the GraphNVP generative model, which is also invertible,  but which does not seem to work very well for sparse low-degree graphs (such as molecular graphs).

The main new idea seems to be to replace the regular coupling flows of GraphNVP with residual blocks: the residual blocks ensure invertibility while also helping to "mix" the information in the (tensor) representations in each layer better. This leads to more compact generative models. Due to the residual structure of the encoding model, the inversion (backward mapping) is a bit more expensive since it requires solving a fixed point problem in each layer, but in principle it can be done provided the layer weight matrices have small spectral norm.

Experimental results on two molecular datasets seem to confirm the validity of the proposed architecture.

The paper is nicely written and the techniques/results are clearly described. However, I have two main concerns:

- Conceptual novelty seems to be limited. The method seems to be a basic modification of the GraphNVP approach by introducing ResNet-like skipped connections. Not entirely sure if this counts as a major contribution.
- Experimental results do not seem to suggest improvements over the state of the art. I am not an expert in molecular generation, but looking at Tables 1 and 2 seem to suggest that the smaller number of parameters in GRF (compared to GraphNVP) come with dramatically reduced performance measures pretty much across the board. This seems like a weak result So I do not know whether the tradeoffs are worth it or not.

Other comments:
- The detailed pseudocode of Algorithm 1 isn't really necessary considering it is just a fixed point iteration.
- Unclear why the reconstruction error in Fig 2 does not monotonically decrease for the ZINC dataset.
- Unclear why Fig 3 suggests smooth variations in the learned latent spaces. I can only spot mode collapses and sudden jumps. It might help to plot VAE/GraphNVP embeddings here too.



**Experience Assessment:**

I have read many papers in this area.

**Review Assessment: Checking Correctness Of Derivations And Theory:**

I assessed the sensibility of the derivations and theory.

**Review Assessment: Checking Correctness Of Experiments:**

I assessed the sensibility of the experiments.

**Review Assessment: Thoroughness In Paper Reading:**

I read the paper at least twice and used my best judgement in assessing the paper.

---

> ### Author Response · Authors · 2019-11-13
> **Response to Reviewer #2**
>
> We thank the reviewer for their comments and questions. We address them in order.
>
> > Conceptual novelty seems to be limited. The method seems to be a basic modification of the GraphNVP approach by introducing ResNet-like skipped connections. Not entirely sure if this counts as a major contribution.
>
> GRF is not merely GraphNVP with skip connections. We construct a flow-based generative model for graphs with residual layers and by computing inverse operation through fixed-point iteration, while GraphNVP adopts affine coupling layers which are analytically invertible.
> Also, it is not obvious that i-ResNet can be applied to graphs, which is discrete and not structured like images. We derive a condition where GRF is invertible reflecting topological information of graphs. From Lemma 1 and 2, the upper bound of the Lipschitz constant of graph residual layers can be decided only on the Lipschitz constant of the activation function (Theorem 1).
>
> > The smaller number of parameters in GRF (compared to GraphNVP) come with dramatically reduced performance measures pretty much across the board.
>
> We observed that increased parameters resulted in instability during training (especially in ZINC-250k). We believe our contribution in terms of its performance is that GRF can achieve comparable scores with less parameters. Thanks to the reduced model size, GRF allows larger batch size than GraphNVP, which is preferable when the number of valid/unique/novel molecules matters rather than their probability.
>
> > Unclear why the reconstruction error in Fig 2 does not monotonically decrease for the ZINC dataset.
>
> We conjecture that the fluctuation after 60 iterations of ZINC-250k is due to the numerical precision of float32 and the larger model size than that for QM9.
>
> > Unclear why Fig 3 suggests smooth variations in the learned latent spaces. I can only spot mode collapses and sudden jumps. It might help to plot VAE/GraphNVP embeddings here too.
>
> First, to the best of our knowledge, there is no standard method that quantitatively evaluates smoothness of the latent space. Thus, we adopted this qualitative evaluation. Though somewhat subjective, this kind of evaluation is common in this area (e.g., JT-VAE (Jin et al., 2018)). We claimed the smoothness of GRF’s learned latent space looking at, for example, the lower right area of Figure 3 (a).
> Mode collapse is a problematic phenomenon of GANs, which does not consider the empirical distribution. On the other hand, flow models in general do not suffer it because they are trained to maximize the likelihood of the whole training data. Some generated samples by GRF are the same because graphs are discrete; two slightly different latent variables can go to the same graphs.
> Sudden jumps can also be attributed to the discreteness. GRF does not guarantee “uniform smoothness.”

---

### Official Review · AnonReviewer1 · 2019-11-01
**Official Blind Review #1**

**Rating:** 3

**Review:**

GraphNVP is the first paper to introduce the concept of "invertible flow", that is to construct the invertible mapping from latent vector z to the graph G. By constructing the mapping from G to z, GraphNVP first changes the discrete feature vector into continuous variables, then update this matrix representation by scaling and transforming functions (Eq. (2)-(5) in this GRF paper). In each iteration the matrix is only updated by one row (one slice for the tensor), while keep other rows intact. Then for constructing the inverse mapping, we can first sample a random vector and then apply the “inverse” of the update rule to recover the edge matrix and node matrix respectively.

The main contribution for GRF paper is to find a new update rule from the idea of ResNet. The author thinks that GraphNVP only update one row each time, which is less efficient, and the model can only cover a limited number of mappings. Then he proposed a new function for update (Eq. (6)-(7)), which updates all rows each time. The author shows how to approximate the determinant of the Jacobian matrix, and how to construct the inverse mapping from fixed-point iteration, as the mapping is Lipschitz. Lastly the author shows that GRF uses much less parameters than GraphNVP both theoretically and practically, which means that the new model is more expressive. However, the new model may output the same molecule for several times (shown by “Uniqueness"), and it favors the striaght-chain molecules.

Another question related to the experiment is that how does the method compared to methods which first generate SMILES string and then convert to molecule graph? Eg. Dai Et al. ICLR 2018, Syntax-directed generative model for structured data.

**Experience Assessment:**

I have published one or two papers in this area.

**Review Assessment: Checking Correctness Of Derivations And Theory:**

I assessed the sensibility of the derivations and theory.

**Review Assessment: Checking Correctness Of Experiments:**

I assessed the sensibility of the experiments.

**Review Assessment: Thoroughness In Paper Reading:**

I read the paper at least twice and used my best judgement in assessing the paper.

---

> ### Author Response · Authors · 2019-11-13
> **Response to Reviewer #1**
>
> We thank the reviewer for their comments and questions. We address them in order.
>
> > The new model may output the same molecule for several times (shown by “Uniqueness"), and it favors the straight-chain molecules.
>
> It is true that our GRF suffers relatively low uniqueness, but GRF generates graphs in one-shot rather than node by node, so it takes less time for generation than autoregression. Compared to GraphNVP, GRF is more memory-efficient and allows larger batches thanks to the substantially reduced model size. If one desires unique molecules, then GRF can repeat graph generation with a larger batch size until the desired number of unique molecules are obtained.
> Also, considering the real-world applications, there are some cases that do not require high uniqueness value. For example, lead optimization can be framed as latent variable optimization towards some molecular property, which is not about generating many unique molecules.
>
> > How does the method compared to methods which first generate SMILES string and then convert to molecule graph?
>
> SMILES-based approaches tend to have low validity and reconstruction, probably because the transformation between molecular structure and SMILES string is very hard. We refer to the Figure 1 of JT-VAE paper (Jin et al., 2018). This example clearly shows that a small difference in the structure can result in a large difference in SMILES. Vice versa.
> Incorporating SMILES syntax improves validity (e.g., Dai et al, 2018), while it involves complicated implementation of decoders/generators. On the other hand, GRF can decode latent variables just by the inverse operation. We can implement the model using modern deep learning frameworks such as PyTorch or TensorFlow.

---

### Decision · Program_Chairs · 2019-12-19

**Decision:**

Reject

**Comment:**

The authors propose a graph residual flow model for molecular generation.  Conceptual novelty is limited since it is simple extension and there isn't much improvement over state of art.